# Porous Chitosan Hydrogels Produced by Physical Crosslinking: Physicochemical, Structural, and Cytotoxic Properties

**DOI:** 10.3390/polym15092203

**Published:** 2023-05-06

**Authors:** Gabriela Fletes-Vargas, Hugo Espinosa-Andrews, José Manuel Cervantes-Uc, Isaías Limón-Rocha, Gabriel Luna-Bárcenas, Milton Vázquez-Lepe, Norma Morales-Hernández, Jorge Armando Jiménez-Ávalos, Dante Guillermo Mejía-Torres, Paris Ramos-Martínez, Rogelio Rodríguez-Rodríguez

**Affiliations:** 1Tecnología de Alimentos, Centro de Investigación y Asistencia en Tecnología y Diseño del Estado de Jalisco A.C (CIATEJ, A.C), Camino Arenero 1227, El Bajío del Arenal, Zapopan 45019, Jalisco, Mexico; lintu_2304@hotmail.com (G.F.-V.); nmorales@ciatej.mx (N.M.-H.); 2Departamento de Ciencias Clínicas, Centro Universitario de los Altos (CUALTOS), Universidad de Guadalajara, Carretera Tepatitlán Yahualica de González Gallo, Tepatitlan de Morelos 47620, Jalisco, Mexico; isaias.limon@cualtos.udg.mx; 3Unidad de Materiales, Centro de Investigación Científica de Yucatán, A.C (CICY A.C), Calle 43 No. 130 X 32 y 34, Chuburná de Hidalgo, Mérida 97205, Yucatan, Mexico; manceruc@cicy.mx; 4Departamento de Polímeros y Biopolímeros, CINVESTAV Unidad Querétaro, Mexico City 76230, Queretaro, Mexico; gabriel.luna@cinvestav.mx; 5Departamento de Ingeniería de Proyectos, Centro Universitario de Ciencias Exactas e Ingeniería (CUCEI), Universidad de Guadalajara, Blvd. Marcelino García Barragán #1421, esq. Calzada Olímpica, Guadalajara 44430, Jalisco, Mexico; 6Departamento de Oncología Celular y Molecular, Centro de Investigación y Desarrollo Oncológico S.A de C.V (CIDO S.A de C.V), San Luis Potosí 78218, San Luis Potosí, Mexico; avalos.joar@gmail.com (J.A.J.-Á.); dantemejia@creo.com.mx (D.G.M.-T.); 7Departamento de Histopatología, Centro de Investigación y Desarrollo Oncológico S.A de C.V (CIDO S.A de C.V), San Luis Potosí 78218, San Luis Potosí, Mexico; 8Departamento de Ciencias Naturales y Exactas, Centro Universitario de los Valles (CUVALLES), Universidad de Guadalajara, Carretera Guadalajara-Ameca Km. 45.5, Ameca 46600, Jalisco, Mexico

**Keywords:** chitosan, physical crosslinking, hydrogels, porous, XPS analysis, porous structure, cytotoxicity

## Abstract

Chitosan hydrogels are biomaterials with excellent potential for biomedical applications. In this study, chitosan hydrogels were prepared at different concentrations and molecular weights by freeze-drying. The chitosan sponges were physically crosslinked using sodium bicarbonate as a crosslinking agent. The X-ray spectroscopy (XPS and XRD diffraction), equilibrium water content, microstructural morphology (confocal microscopy), rheological properties (temperature sweep test), and cytotoxicity of the chitosan hydrogels (MTT assay) were investigated. XPS analysis confirmed that the chitosan hydrogels obtained were physically crosslinked using sodium bicarbonate. The chitosan samples displayed a semi-crystalline nature and a highly porous structure with mean pore size between 115.7 ± 20.5 and 156.3 ± 21.8 µm. In addition, the chitosan hydrogels exhibited high water absorption, showing equilibrium water content values from 23 to 30 times their mass in PBS buffer and high thermal stability from 5 to 60 °C. Also, chitosan hydrogels were non-cytotoxic, obtaining cell viability values ≥ 100% for the HT29 cells. Thus, physically crosslinked chitosan hydrogels can be great candidates as biomaterials for biomedical applications.

## 1. Introduction

Hydrogels are exciting biocompatible biomaterials widely used in biomedical applications due to their exceptional properties, such as biocompatibility, biodegradability, non-toxicity, and similarity to native tissue [1,2]. The hydrogels are hydrophilic biomaterials characterized by a three-dimensional porous structure and cross-linked polymer networks. The hydrogels can absorb and retain substantial amounts of water and biological fluids without disintegrating or losing their structure [1,3]. For example, the large quantity of fluids that a hydrogel can absorb helps to maintain a moist atmosphere on the wound, which promotes the healing process [4]. A balance between cohesive and osmotic forces carries out the water absorption of the hydrogel. It depends on the chemical nature of the functional groups, cross-linking, porosity degree, and hydrogel pore size [5]. Also, hydrogels can regulate drug release through swelling, diffusion, and biodegradation mechanisms, which can occur at different phases of the drug delivery process. If the drug is hydrophilic, a diffusion process will carry out the release. In contrast, if the drug is hydrophobic, the release will be carried out by a degradation process [6,7].

Generally, the hydrogels are fabricated with natural polymers (e.g., chitosan, cellulose, gelatin, pectin, chondroitin sulfate, or collagen, etc.) and synthetic polymers (e.g., polyvinyl alcohol, polycaprolactone, or polylactic-co-glycolic acid, etc.) [8]. Chitosan is the second most abundant polymer after cellulose and the more widely used to produce hydrogels. Chitosan is a polysaccharide obtained by alkaline or enzymatic hydrolysis of chitin (deacetylation process), the main component of crustacean shells such as shrimp, crabs, and lobsters, as well as insects and in the cell walls of fungi [9,10,11]. The chitosan structure comprises two units randomly distributed within their polymer structure: 2-acetamido-2-deoxy-β-D-glucosamine units and 2-amino-2-deoxy-β-D-glucosamine units [1,12]. Chitosan is soluble in acidic aqueous media (e.g., hydrochloric acid, acetic, etc.), in which protonation of the amino groups attached to its polymer structure avoids the formation of interchain interactions through electrostatic repulsion (hydrophobic junctions and hydrogen bonding) [13]. Chitosan is suitable for biomedical applications due to its non-toxicity, biocompatibility, biodegradability, antibacterial, antioxidant, anticancer, anti-inflammatory, hemocompatible, and hemostatic properties [11,14]. In addition, Chitosan has demonstrated functional properties such as muco-adhesion, transfection, in situ gelation, permeation improvement, colon targeting, and efflux pump inhibition. These properties are due to the presence of the NH_2_ groups on the chitosan structure [15,16].

Chitosan solution can be tailored in hydrogels, beads, membranes, microparticles, sponges, and fibers [17]. For example., physical and chemical crosslinking methods can produce chitosan hydrogels [7,18,19]. Chemical processes are produced by a chemical reaction between chitosan and cross-linkers molecules such as glutaraldehyde, glyoxal, genipin, or oxidized dextran, which form covalent bonds among the polymer chains [7,20]. However, chemical cross-linkers can cause toxicity [7,21]. The applications of chitosan hydrogels are specially linked to the microstructure, rheological properties, biocompatibility, and capacity to regenerate biological tissues.

Recently, there has been increasing interest in producing physical hydrogels without chemical cross-linkers due to their increased safety and lower cost [22]. Physical methods include ionotropic gelation (sodium tripolyphosphate), thermogelation (β-glycerophosphate and sodium bicarbonate), and neutralization processes [1,23]. The physical crosslinking by neutralization consists of direct and controlled exposure to alkaline solutions such as sodium hydroxide, urea, gaseous ammonia, sodium bicarbonate, and sodium carbonate, in which non-covalent bonds (hydrophobic and hydrogen bonding) are formed [1,20,24,25]. The neutralization process encourages a loss of charge of chitosan, caused by an increase of the pH near the *pKa* value of the NH_3_^+^ groups, favoring flexibility between the polymer chains and the modification of the hydrophobic/hydrophilic interactions to induce gelation [13]. For example, Enache et al. [17] evaluated the coagulation kinetics of chitosan at different chitosan and sodium hydroxide (NaOH) concentrations. The authors reported that the chitosan hydrogels displayed nonhomogeneous morphology during coagulation kinetic, faster at low and high NaOH concentrations. Reddy et al. [26] produced chitosan hydrogels at different chitosan concentrations using NaOH as a physical crosslinking agent. The authors obtained weak and fluffy scaffolds at low chitosan concentration (1%), while rigid and stronger scaffolds were obtained at high chitosan concentration (5%).

Moreover, Xu et al. [27] described that strong bases like NaOH are associated with the presence of residues in the polymeric structure of hydrogels, reducing their biocompatibility. In this sense, NaOH is a strong base that can damage bioactive molecules, cells, or drugs loaded in hydrogels. Therefore, coagulation conditions using strong bases could decrease the applicability of chitosan-based hydrogels in biomedical applications. Previously, Bergonzi et al. [24] developed physically crosslinked chitosan hydrogels using strong and weak bases such as KOH, Na_2_CO_3_, NaHCO_3_, and ammonia. However, the chitosan hydrogels obtained by Na_2_CO_3_ and NaHCO_3_ as crosslinking agents did not conserve a stable and 3D polymeric structure. Ernesto et al. [28] developed chitosan dressings by a freeze-drying process and sodium bicarbonate as a physical crosslinking agent. The chitosan samples displayed interconnected porous structures with a fluid’s absorption capacity above 900%, suitable biocompatibility, cell adhesion, and proliferation of fibroblasts. However, the authors only evaluated these properties at a single chitosan concentration (2% wt.) and chitosan molecular weight (147,000 g/mol). Recently, our research group produced novel physical chitosan hydrogels at different chitosan concentrations (1.5 and 2% wt.) and molecular weights (101,281 g/mol and 215,954 g/mol) using sodium bicarbonate as a physical crosslinking agent [29]. The molecular weight and chitosan concentration influenced the hydrophilicity, degradation temperature, pore size (131–239 µm), and viscoelastic properties of chitosan hydrogels. However, no studies have investigated water absorption, internal microstructure in the hydrated state, and in vitro cytotoxic studies of physical chitosan hydrogels using sodium bicarbonate as a crosslinking agent at different chitosan concentrations and molecular weights, which are essential properties for biomedical applications.

The purpose of this study was to evaluate the chemical identity and elemental composition (XPS), equilibrium water content, microstructural morphology in the hydrated state (confocal microscopy), mechanical properties (temperature sweep test), and in vitro cytotoxicity (MTT assay) of physically crosslinked chitosan hydrogels by sodium bicarbonate at different chitosan concentrations and molecular weights. These hydrogels could be used as potential biomaterials (dressing or scaffolds) for biomedical applications.

## 2. Materials and Methods

### 2.1. Materials

Chitosans of low molecular weight (101,281 g/mol) and medium molecular weight (215,954 g/mol) were supplied from Sigma Aldrich (St. Louis, MO, USA). All other chemicals were analytical (Sigma Aldrich, St. Louis, MO, USA).

### 2.2. Fabrication of Hydrogels

Firstly, both chitosan powders were dispersed in an acetic acid solution (2% *w*/*v* acid solution) and gently stirred for 24 h at 25 °C. Next, we prepared two chitosan concentrations (1.5 and 2.0 wt %) from each chitosan molecular weight. Then, approximately 8 g of each chitosan solution was put into a plastic petri dish (14 mL) and frozen at −20 °C for 24 h. The samples were freeze-dried at −53 °C and 0.016 mBar for 72 h (LABCONCO model 4.5 FreeZone, Kansas City, MO, USA). Next, the dried samples were immersed in a NaHCO_3_ solution (0.2 M) for 30 min at 20 °C to allow the hydrogel formation. Then, chitosan hydrogels were washed several times with a PBS buffer. Finally, the chitosan hydrogels were sterilized at 121 °C and 103.4 kPa for 15 min using an autoclave SE 510 (Figure 1) (Yamato Scientific, Tokyo, Japan). The chitosan hydrogels produced were labeled like: (i) Sample BC-H1 (1.5% *w/v*; low molecular weight), (ii) Sample BC-H2 (1.5% *w/v*; medium molecular weight), (iii) Sample BC-H3 (2.0% *w/v*; low molecular weight), and (iv) Sample BC-H4 (2.0% *w/v*; medium molecular weight).

The chitosan hydrogels were freeze-dried for X-ray photoelectron spectroscopy, XRD, and equilibrium water analysis (dried chitosan samples). Immediately, the dried chitosan samples were stored in a desiccator with phosphorus pentoxide for two weeks to ensure total water removal.

### 2.3. Physicochemical Characterization of Chitosan Hydrogels

#### 2.3.1. X-ray Photoelectron Spectroscopy

X-ray photoelectron spectroscopy (XPS) analysis was carried out using the Alpha110 instrument from Thermo Fisher Scientific (Waltham, MA, USA) with a monochromatic Al Ka1 (1486.7 eV) X-ray source and a hemispherical electron analyzer with seven channeltrons. The peaks fitted with the software AAnalyzer^®^ v. 1.1. The calibration of the binding energy of the spectra was performed with the C1s peak of the carbon due to CAC bonding at 284.8 eV. The survey spectra were used to quantify the composition of the samples, calculate the relative atomic composition, and detect any impurities that might be present. Data were analyzed using CasaXPS software (Casa Software, Teignmouth, UK).

#### 2.3.2. XRD Analysis

X-ray diffraction measurements of the different dried chitosan samples were acquired by an X-ray diffractometer Dmax2100 (Rigaku, Tokyo, Japan) having a CuKα (λ = 0.154 nm) at a scanning speed of 10° per min over a 2θ range of 5–40° [8].

#### 2.3.3. Morphological Properties

The internal microstructure of the chitosan hydrogels was observed by confocal microscopy DMRA2 (Leica Microsystems GmbH, Wetzlar, Germany) equipped with an Evolution QEi camera (MediaCybernetics, Bethesda, MD, USA). The chitosan hydrogels were labeled with rhodamine B isothiocyanate in methanol (0.1 mg/mL) for 24 h. The residual rhodamine was removed from the samples using PBS buffer. The internal structure of the chitosan hydrogels was obtained using excitation/emission wavelengths of 530/590 and observed with 100× magnification [8].

#### 2.3.4. Equilibrium Water Content

The dried chitosan samples were immersed in a PBS buffer solution at 37 °C for 24 h. After removing the hydrogels, the excess PBS buffer was drained, and the samples were weighed. The equilibrium water content (%) of the chitosan hydrogels was calculated [30]:(1)$$EWC\left(\%\right)=\frac{W_{s}-W_{d}}{W_{d}}\times \;100\%$$
where *W_s_* is the mass of swollen hydrogel after 24 h hydration, and *W_d_* is the mass of the dry sample before submersion.

#### 2.3.5. Rheological Properties

The effect of the temperature on the viscoelastic properties (elastic and viscous modulus) of the chitosan hydrogels was evaluated using an AR1000 rheometer (TA Instruments, Newcastle, DE, USA). A cross-hatched parallel plate geometry of 40 mm was used. The gap between the flat surfaces was set at 2.5 mm. Temperature sweep test evaluated changes in the rheological modulus of the hydrogels from 5 to 60 °C at a heating rate of 5 °C/min. Constant oscillation frequency and strain of 6.23 rad/s and 1% were used [30].

### 2.4. In Vitro Cytotoxicity

In vitro, cytotoxicity of chitosan hydrogel extracts was carried out by the extraction method (indirect method) according to the guideline of international standard ISO 10993-5:2009 using HT29 human colorectal adenocarcinoma cells (HTB38 ATTC) [8,31]. The cells were cultured in Dulbecco’s modified Eagle’s medium (DMEM, Sigma Aldrich) supplemented with 10% fetal bovine serum (Sigma Aldrich). HT29 cells were incubated at 37 °C in a humidified atmosphere of 5% CO_2_. After reaching 85% confluency, the cells were detached by trypsinization, seeded, and incubated in a 96-well cell culture plate at 5 × 10^4^ cells per well for 24 h (37 °C and 5% CO_2_). After this time, the cell culture media was replaced with the extract media at different concentrations (0, 12.5, 25, 50, and 100%).

#### Cytotoxicity of the Hydrogel Extract

The cytotoxicity of the chitosan hydrogels was evaluated using the MTT (3-[4,[5-dimethylthiazol-2-yl]-2,5-diphenyl tetrazolium bromide) assay [8,30]. First, sterilised hydrogels were incubated with DMEM serum-free medium at 37 °C for 48 h. First, the hydrogels were isolated from the extract media, and the latter was supplemented with fetal bovine serum (10%). Next, the extracts media were diluted with fresh DMEM medium to obtain extracts with a concentration of 12.5%, 25%, 50%, and 100%. Cells cultivated in supplemented DMEM medium without extract were used as the negative control, while DMSO (10%) was used as a positive control. Then, 100 μL of the extracts at different concentrations were added to a 96-well plate coated with cells and incubated at 37 °C for 24 h. Later, 100 μL MTT (0.5 mg/mL in DMEM medium) was added into each well and incubated for four hours at 37 °C. Finally, the media was removed, and 200 μL of dimethyl sulfoxide was added to each well to dissolve formazan crystals. The absorbance was measured using a microplate reader (TECAN infinite PRO-200, Trading AG, Steinhausen, Switzerland) at a wavelength of 570 nm. The following equation calculated the cell viability:(2)$$\%\;cell\;viability=\frac{I_{extract}}{I_{control}}\times \;100\%$$
where, $I_{extract}$ is the absorbance of the cells exposed to the extract and $I_{control}$ is the absorbance of the cells incubated without extract.

### 2.5. Statistical Analysis

All measurements were in triplicate and reported as means ± standard deviations. ANOVA analysis was applied, and Tukey’s test evaluated differences among the means (*p* < 0.05). Statgraphics Centurion XVI program version 16.1.17 (StatPoint Technologies, Inc., Warrenton, VA, USA) was used for the statistical analyses.

## 3. Results and Discussion

In this research study, physically crosslinked chitosan hydrogels were successfully produced and physicochemically (X-ray spectroscopy, equilibrium water content, microstructural morphology, rheological properties) and biologically (cytotoxicity cell) characterized.

### 3.1. XRD Spectroscopy

Figure 2 shows the X-ray diffraction patterns of dry uncross-linked chitosan samples (chitosan acetate samples) and dried chitosan hydrogels physically cross-linked with sodium bicarbonate. The XRD patterns of the dry uncross-linked chitosan hydrogels (UC-H1 and UC-H4 samples) showed a broad peak, while UC-H2 and UC-H3 displayed a weak peak, indicating a semi-crystalline structure [32,33]. The central peak of these samples was located at 2θ = 20.9°, a typical chitosan peak (Figure 2a). Reddy et al. [26] reported a similar peak in bulk chitosan. In our study, the high-intensity peaks corresponding to interplanar distances of 4.436 Å were observed in samples UC-H1 and UC-H4. Li et al. [22] and Alghuwainem et al. [34] attributed these peaks to the strong intramolecular hydrogen bonds among the chitosan polymeric chains.

Figure 2b revealed changes in the semi-crystalline structure induced by physical cross-linking with sodium bicarbonate (BC-H1, BC-H2, BC-H3, and BC-H4). The diffraction peak centered at 2θ = 20° of BC-H1, BC-H3, and BC-H4 samples displayed higher intensity than the BC-H2 sample. These results are related to the formation of crystalline regions due to the physical crosslinking between polymeric chains (entanglement and hydrogen bonds). Chitosan is a partially crystalline polysaccharide with some crystalline forms fixed in the amorphous region. For example, BC-H2 was produced at 1.5% of chitosan medium molecular weight. In this condition, the XRD peak at 2θ = 20° diminished due to the increase of the amorphous region by the formation of new polymeric networks and possible changes in the internal microstructure of the chitosan hydrogels upon physical crosslinking networks [35].

In addition, the crosslinked chitosan samples showed other interesting diffraction peaks: a low-intensity peak at 2θ = 9.6°, as well as four intense peaks at 2θ = 27.2, 31.6, 45.4°, and 56.3°. These peaks are attributed to residual sodium chloride crystals [8,36]. Previously, the cross-linked chitosan samples were sterilized in PBS buffer, so residual sodium chloride salts could be present. Reddy et al. [26] reported a significant decrease in the intensity of the characteristic peak of chitosan after crosslinking with NaOH. The authors attributed these results to the hydrolysis and damage to the structure of chitosan. Thus, sodium bicarbonate can offer a treatment safe to produce chitosan hydrogels without damaging its polymeric structure.

### 3.2. X-ray Photoelectron Spectroscopy

The XPS analysis evaluated the structure and elemental composition of dry uncross-linked chitosan samples and dry cross-linked chitosan hydrogels (Figure 3). The XPS spectrum of dry uncross-linked chitosan samples (Figure 3a) showed two intense peaks at 282.8 and 535.6 eV, corresponding to the binding energy of C 1s (~71%) and O 1s (~22%), respectively. Two weak peaks were also observed at 344.7 and 401.3 eV, corresponding to the binding energy of Ca 2p (0.2%) and N 1s (~5%), respectively.

The XPS spectrum of dried cross-linked chitosan hydrogels shows three new peaks at 129.5, 196.5, and 1072.4 eV, which are attributed to elements such as P 2p (~1%), Cl 2p (~10%), and Na 1s (~5%), respectively (Figure 3b). The presence of these elements is attributed to the PBS buffer used during the sterilization process (Section 2.2), which remained within the polymeric structure of the chitosan hydrogels. These results agree with the above XRD analysis (Figure 2b).

Figure 4 shows the high-resolution N 1s XPS spectra of dried uncross-linked chitosan samples (Figure 4a,b) and dried cross-linked chitosan hydrogels (Figure 4c–f).

The N 1s peaks of the dried uncross-linked chitosan samples were resolved with two Gaussian peaks with different bonding states separated into two electronic states (Figure 4a,b): (i) amine groups at 398.8 and 399.7 eV (-C-NH_2_) and (ii) secondary amide groups (-HN-C = O) at 399.9 and 401 eV for UC-H1 and UC-H2, respectively [22,37]. The N 1s peaks of the dried cross-linked chitosan hydrogels shifted to lower binding energy values. In contrast, the relative peak area attributed to the amine groups decreased by the physical crosslinking of the chitosan samples. For example, the binding energy values of the amine groups (-C-NH_2_) were 398.6, 397.1 398.1, and 397.7 eV for BC-H1, BC-H2, BC-H3, and BC-H4, respectively.

### 3.3. Equilibrium Water Content

The equilibrium water content is a critical property of hydrogels for biomedical applications. It is related to their ability to absorb body fluids and transfer nutrients and metabolites through the polymeric structure [30,38]. Figure 5 shows the equilibrium water content of the chitosan hydrogels under physiological conditions (PBS buffer, pH 7.4).

The chitosan hydrogels absorbed high buffer PBS concentration, with equilibrium water content values of 23 to 30 times their mass after 24 h immersion: i.e., 3049.2 ± 36.5, 2505.67 ± 76.9, 2773.2 ± 85.7, and 2312.9 ± 91.3% for BC-H1, BC-H2, BC-H3, and BC-H4, respectively. This behavior is attributed to the high hydrophilicity of chitosan and the porosity of the hydrogels. Begum et al. [39] produced chitosan scaffolds using an aqueous solution of NaOH/ethanol as a gelling agent. The chitosan hydrogels absorbed a large amount of PBS buffer, about 60 times their initial mass while adding alginate reduced their swelling capacity to 30 times. The authors attributed these results to the increased rigidity and strength of the scaffolds with the addition of alginate, allowing them to absorb PBS without swelling.

The results obtained in our study displayed that the concentration and molecular weight of the chitosan hydrogels influenced the equilibrium water content capacity of chitosan hydrogels (*p* < 0.05). The equilibrium water content decreased with increasing chitosan molecular weight. The swelling properties depended on the presence of hydrophilic groups attached to the polymer backbone (amino and hydroxyl groups). Kaczmarek et al. [40] described that high water uptake values after incubation were associated with the water flow through the polymer structure and the pores filling. The equilibrium water content values for BC-H2 and BC-H4 chitosan hydrogels were significantly lower (*p* < 0.05) than those for BC-H1 and BC-H3 chitosan hydrogels. The lower equilibrium water contents of the BC-H2 and BC-H4 chitosan hydrogels were related to the loss of charge density of the glucosamine segments (*pKa* = ~6.3–7) of chitosan, which decreased the hydrogen bonding with water molecules. Seda-Tığlı et al. [41] evaluated the effect of chitosan concentration (2 and 3% wt.) and deacetylation degree (DDA, 75–85%, and >85%) on the swelling behavior (PBS buffer) of porous chitosan hydrogels physically crosslinked by ethanol. The chitosan hydrogels displayed an equilibrium water content between 27 and 40 times their initial mass, displaying chitosan concentration and deacetylation degree with a negative and positive effect, respectively. Moreover, Ernesto et al. [28] reported an equilibrium water content value of 886% (PBS buffer) and 1072% (culture medium) for physical chitosan hydrogels using sodium bicarbonate as a crosslinking agent. In this sense, our results demonstrated that chitosan hydrogels at different chitosan concentrations and molecular weights possess higher swelling capacity than those previously reported.

### 3.4. Viscoelastic Properties

Viscoelastic properties are a practical methodology for studying hydrogels. Hydrogels are tridimensional networks with viscoelastic behavior (viscous and elastic behavior). Typically, the elastic modulus (G’) is measured to understand hydrogels’ microstructure, heterogeneity, and crosslinked state. The temperature stability of the hydrogels is a desirable property for biomedical applications. For example, if a hydrogel shows stability against temperature, it could be a potential candidate as a scaffold for tissue engineering. Instead, if the hydrogel shows low stability in the temperature range between 30 to 40 °C could be a good candidate as a biodegradable medical implant. The oscillatory rheology test evaluated the changes in the viscoelastic properties of chitosan hydrogels as a function of temperature (Figure 6).

The elastic modulus of the chitosan hydrogels was higher than the loss modulus within the temperature range due to the formation of robust physical entanglement polymeric networks. At 37 °C, the BC-H1, BC-H2, BC-H3, and BC-H4 displayed elastic modulus values of 2196, 2376.5, 3414, and 5301 Pa, respectively. The BC-H4 hydrogel was approximately 2.3 times folder than the BC-H1 and BC-H2 chitosan hydrogels and almost 1.5 times higher than the BC-H3 hydrogel at a constant temperature. If the elastic modulus is high, the hydrogel is more deformation-resistant and requires more stress to be deformed. In addition, the chitosan hydrogels showed high stability to temperature changes from 5 to 60 °C. For example, the elastic modulus changes in the temperature range from 30 to 40 °C were 3.3, 7.0, 6.4, and 5.8% for BC-H1, BC-H2, BC-H3, and BC-H4, respectively. These results suggest that the chitosan hydrogels can be used as scaffolds in biomaterial for biomedical applications. Li et al. [22] reported similar behavior in chitosan hydrogels physically crosslinked by ammonia. The chitosan hydrogels showed a slight decrease in elastic modulus with increasing temperature. Also, the authors found that the elastic modulus of the hydrogels decreased considerably after 60 °C.

The mechanical stability of hydrogels is a critical parameter that affects their biomedical application (e.g., tissue engineering). Hydrogel as a scaffold help to preserve and supports the cell/tissue structure, maintaining cell differentiation and proliferation. Also, scaffolds should offer mechanical support to the applied loading during in vivo implantation and in vitro cell culture [39,42].

### 3.5. Microstructure by Confocal Microscopy

Freeze-drying is a process generally used to produce highly porous scaffolds. During freezing, ice crystals nucleate from the polymeric solution and grow along thermal gradients, leading to various pore sizes according to freezing temperatures. Therefore, pore size can be controlled by the ice crystal size by varying the freezing temperature and freezing rate [28,43].

In our study, confocal microscopy was used to evaluate the internal microstructure of the chitosan hydrogels in the hydrated state. The porous structure of the chitosan hydrogel should offer a suitable template for cells to grow in and seed new tissue regeneration [23]. Unlike conventional SEM, confocal microscopy allows observing the internal microstructure of the swollen polymeric structure. Figure 7 shows the internal polymeric network of chitosan hydrogels in a swollen state at different chitosan concentrations and molecular weights labeled with rhodamine B.

The chitosan hydrogels showed an integral polymeric structure without collapse, presenting irregular porous polymer structures of different pore sizes. The mean pore sizes of the chitosan hydrogels were calculated from confocal micrographs. The BC-H1, BC-H2, BC-H3, and BC-H4 hydrogels showed mean pore size values of 120.6 ± 22.6, 156.3 ± 21.8, 119.2 ± 19.6, and 115.7 ± 20.5 µm, respectively. In this sense, the polymer concentration and molecular weight of chitosan did not affect the apparent mean pore sizes in the chitosan hydrogels (*p* < 0.05). The polymer structure expands as the hydrogel hydrates, increasing the pore size. Tolstova et al. [44] reported similar mean pore size (150 ± 5 μm) for chitosan hydrogels produced by the freeze-drying and thermal crosslinking process (150 °C for 5 h). The authors used fluorescamine as amino-specific staining to observe the macrostructure of chitosan hydrogels.

The porous microstructure and appropriate pore size of hydrogels would help cells to adhere and spread appropriately on the surface of hydrogels [45]. For example, Azueta-Aguayo et al. [25] described that pore sizes of about 100 µm are suitable for cell proliferation. In addition, the interconnected porous structures, and the high degree of porosity of the scaffolds can provide good nutrient and oxygen transfer, resulting in improved cell migration and ECM production inside the scaffolds [8,44].

### 3.6. In Vitro Cytotoxicity

Cytotoxicity is one of the most critical properties of biomaterials in biomedical applications. The MTT assay evaluated the cellular metabolic activity by the indirect cytotoxic response to physically crosslinked chitosan hydrogels [8]. Figure 8 displays the cell viability of HT-29 at different extract concentrations of chitosan hydrogels after an incubation time of 24 h at 37 °C.

The cell viability of the extracts 12.5% (samples BC-H1 and BC-H3), 25% (BC-H1, BC-H2, BC-H3, and BC-H4), 50% (BC-H1 and BC-H4), and 100% (BC-H2, BC-H3, and BC-H4) showed statistical differences (*p* < 0.05) after 24 h of incubation compared to the negative control (–CTRL), reaching cell viability values higher than 100%. The normative ISO 10993-5:2009 describe that an extract is considered cytotoxic if the cell integrity is compromised (more than 30% cell death). These results indicate that it did not show toxic potential. The chemical structure of chitosan is similar to hyaluronic acid, which confers high biocompatibility for developing different scaffolds for biomedical applications [46]. The results suggest that chitosan hydrogels could promote the proliferation of cells due to excellent cytocompatibility. Moreover, HT-29 cells exposed to 12.5% (BC-H2 and BC-H4), 50% (BC-H2 and BC-H3), and 100% (BC-H1) maintained the cell metabolic activity after 24 h of incubation (*p* < 0.05). The results were similar to those reported by Rodríguez-Rodríguez et al. [30] on chitosan-based hydrogels. Hence, the physically crosslinked chitosan hydrogels would possess suitable biocompatibility attributed to their synthesis method.

## 4. Conclusions

This study successfully produced chitosan hydrogels using sodium bicarbonate as a crosslinking agent at different concentrations and molecular weights. The freeze-drying process was an effective method to produce porous with numerous cavities within its polymer structure. The XPS analysis confirmed the physical crosslinking of the chitosan hydrogels. Chitosan hydrogels’ high hydrophilicity and elastic behavior can provide suitable conditions for cell adhesion and growth. Furthermore, the physically crosslinked chitosan hydrogels were biocompatible, displaying non-cytotoxicity for HT-29 cells. Thus, physically crosslinked chitosan hydrogels at different concentrations and molecular weights can be great candidates as biocompatible polymeric scaffolds and dressings for biomedical applications.

## Figures and Tables

**Figure 1 polymers-15-02203-f001:**
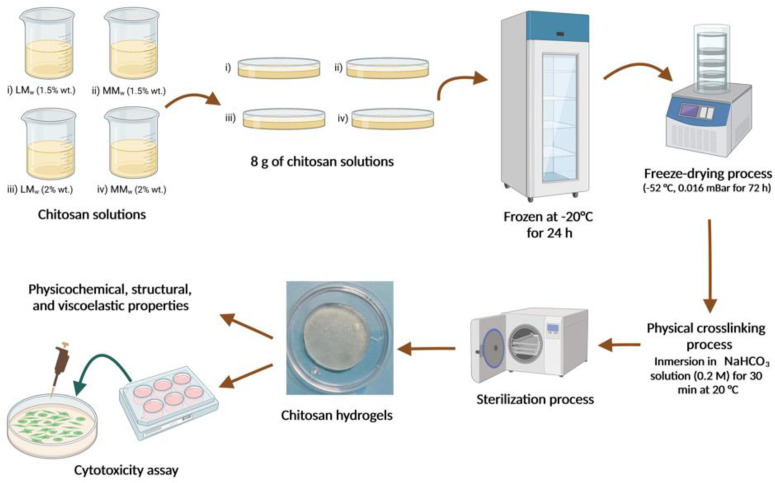
Scheme of preparation of physical chitosan hydrogels using sodium bicarbonate as a crosslinking agent. Created with BioRender.com (Access date 1 May 2023).

**Figure 2 polymers-15-02203-f002:**
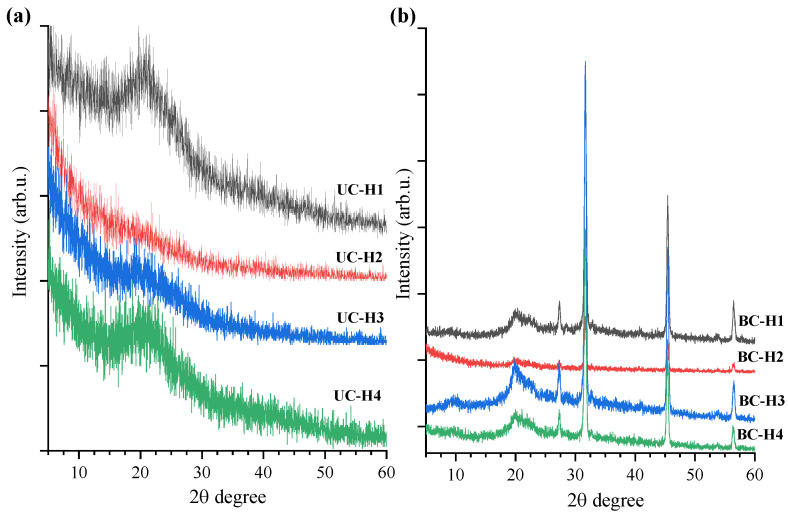
XRD diffraction of (**a**) dried uncross-linked chitosan hydrogels (UC-H1, UC-H2, UC-H3, and UC-H4) and (**b**) dried crosslinked chitosan hydrogels (BC-H1, BC-H2, BC-H3, and BC-H4).

**Figure 3 polymers-15-02203-f003:**
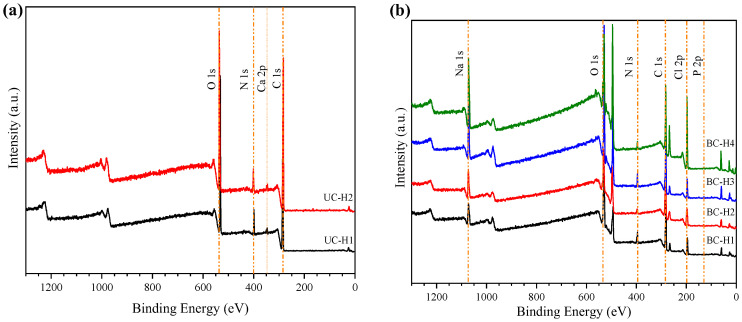
XPS spectra of (**a**) dried uncross-linked chitosan samples (UC-H1 and UC-H2) and (**b**) dried cross-linked chitosan samples (BC-H1, BC-H2, BC-H3, and BC-H4).

**Figure 4 polymers-15-02203-f004:**
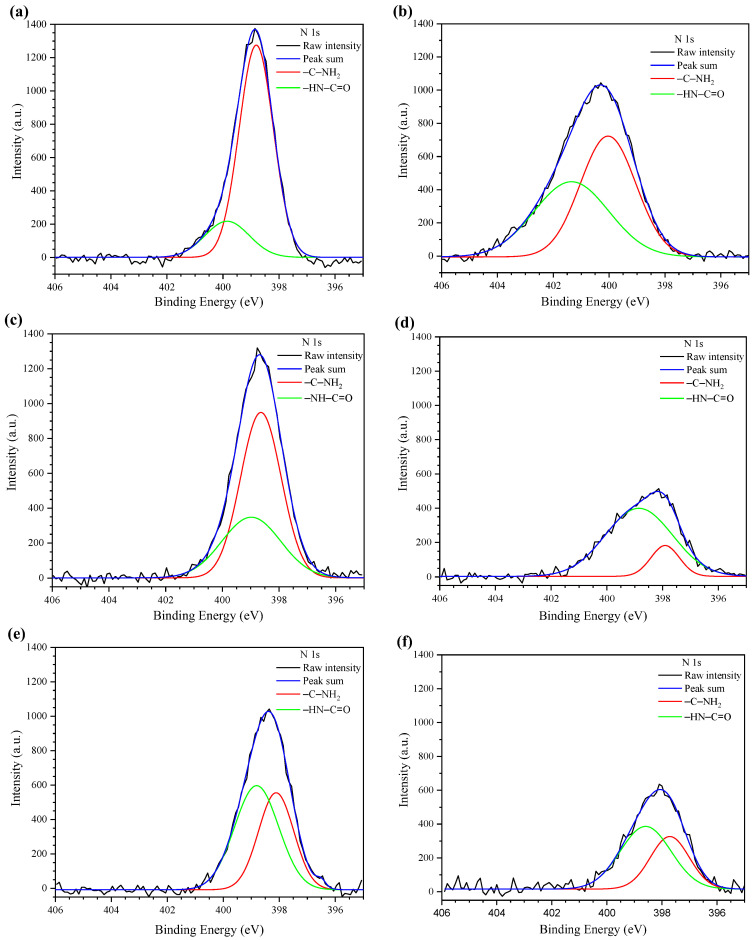
N1s XPS spectra of dried uncross-linked chitosan samples (**a**) UC-H1 and (**b**) UC-H2 and dried cross-linked chitosan samples (**c**) BC-H1, (**d**) BC-H2, (**e**) BC-H3, and (**f**) BC-H4.

**Figure 5 polymers-15-02203-f005:**
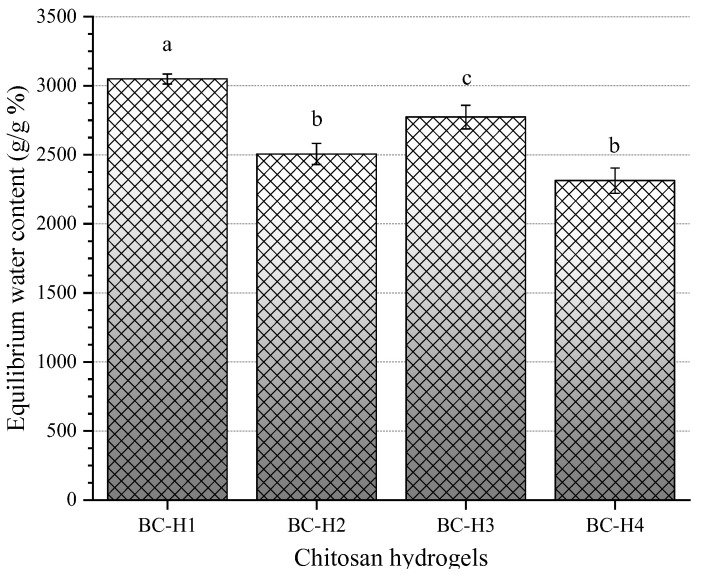
The equilibrium water content of physically crosslinked chitosan samples in PBS buffer solution (pH 7.4). Mean ± SD, n = 3, ^a,b,c^ same letter are considered not significantly different according to Tukey HSD test (*p* < 0.05).

**Figure 6 polymers-15-02203-f006:**
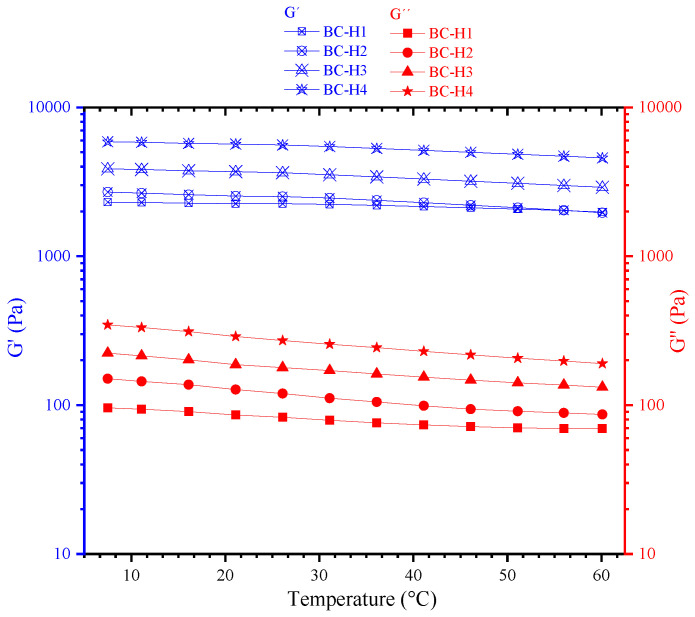
Temperature sweeps of physically crosslinked chitosan hydrogels: Hydrogels [BC-H1] (square), [BC-H2] (circle), [BC-H3] (triangle), and [BC-H4] (star). Values of G’ (empty cross blue symbols), G’’ (red filled symbols).

**Figure 7 polymers-15-02203-f007:**
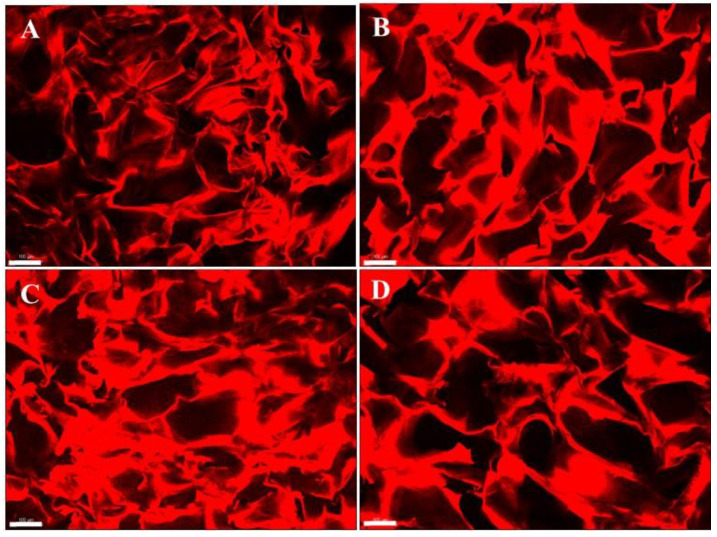
Confocal micrographs of physically cross-linked chitosan hydrogels: (**A**) (hydrogel BC-H1), (**B**) (hydrogel BC-H2), (**C**) (hydrogel BC-H3), and (**D**) (hydrogel BC-H4) (scale bar: 100 μm). Chitosan hydrogels were treated with rhodamine B. Red fluorescence evidenced the polymer structure of the hydrogels.

**Figure 8 polymers-15-02203-f008:**
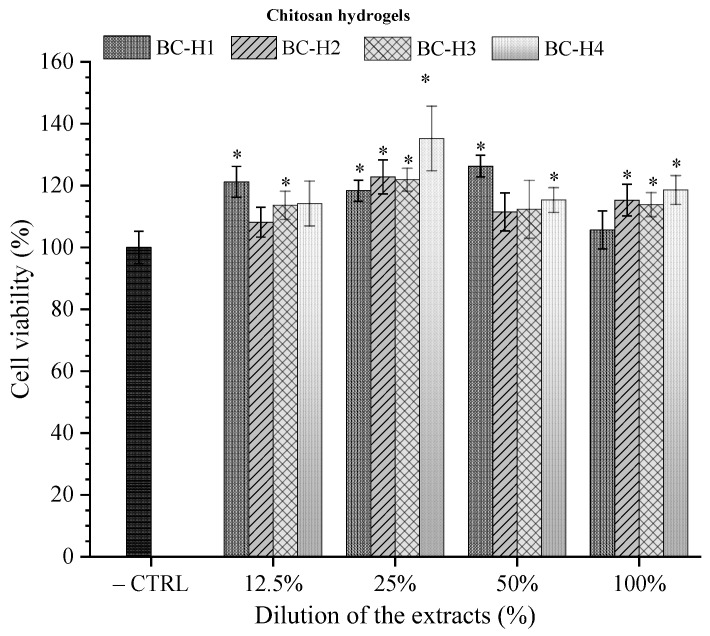
In vitro cytotoxicity of the extract media chitosan hydrogels toward HT-29 cells by MTT assay. * Represents a significant difference compared to the negative control (cells cultivated in DMEM medium without extract; —CTRL) (*p* < 0.05).

## Data Availability

Data will be made available on request.

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
