# Peer review of "Porous Chitosan Hydrogels Produced by Physical Crosslinking: Physicochemical, Structural, and Cytotoxic Properties"

_polymers, 2023, doi:10.3390/polym15092203_

Round 1

Reviewer 1 Report

The article presented can be of interest to the readers of the journal.

But there are some questions and notes to arise while reading the manuscript.

1. The authors have not explained what like look their hydrogels in the crosslinked state. Are these materials the viscous liquids, the quasi-solid matters (as, for example, the PAAM or agar gels), or something else?

2. The authors state that “The equilibrium water content is a critical property of hydrogels for biomedical applications”. But some questions can be put forward while reading the paragraph describing the results of the estimation of this value. The authors do not present any proofs of full removal of water from the material during its pre-drying. So the correctness of the equilibrium water content measured in the work could provoke some hesitations.

3. The figures 1-5 are of the awfully poor quality! This circumstance does not permit to evaluate correctly the data presented in the manuscript.

4. The critically important feature of the hydrogels studied is their crosslinked state. So it would be important to try to evaluate the crosslinks density values inherent to the materials obtained. Do these densities differ significantly for the four materials? How the authors have optimized the protocol of the crosslinking process? Could the crosslinks density values realized in the work be treated as optimal for practical use of the materials studied, or they need some further tuning?

5. The data presented show the substantial difference of the properties of crosslinked gel H2 and all other gels studied. H2 demonstrates very little intensity of diffraction peak centered at 2θ = 20° inherent to chitosan (Fig. 1b), and on the other hand is characterized by substantially larger pore size as compared to all other materials. Could the authors put forward some explanation of the “unusual” properties of this H2 hydrogel?

6. It would be reasonable to point out in the “Conclusions” section which of the four materials under study is more suitable for the practical use in biomedical applications.

Author Response

Reviewer 1.

  1. The article presented can be of interest to the readers of the journal. But there are some questions and notes to arise while reading the manuscript.

Reply: Thank you for your comments.

  1. The authors have not explained what like look their hydrogels in the crosslinked state. Are these materials the viscous liquids, the quasi-solid matters (as, for example, the PAAM or agar gels), or something else?

Reply: Thank you for your comments. The physically crosslinked chitosan hydrogels displayed an elastic structure, i.e., they behave like a solid-viscoelastic. Also, the elastic behavior of the hydrogels was conserved throughout the temperature range (5-60 °C) due to the formation of robust physical entanglement polymeric networks.

  1. The authors state that “The equilibrium water content is a critical property of hydrogels for biomedical applications”. But some questions can be put forward while reading the paragraph describing the results of the estimation of this value. The authors do not present any proofs of full removal of water from the material during its pre-drying. So the correctness of the equilibrium water content measured in the work could provoke some hesitations.

Reply: We apologize for the misunderstanding. Once the chitosan hydrogels were produced, they were dehydrated by a freeze-drying process. Immediately, the dried chitosan samples were stored in a desiccator with phosphorus pentoxide for two weeks to ensure total water removal. This salt is used to decrease water activity near zero.

  1. The figures 1-5 are of the awfully poor quality! This circumstance does not permit to evaluate correctly the data presented in the manuscript.

Reply: We apologize for the inconvenience. The quality of the Figures was improved.

  1. The critically important feature of the hydrogels studied is their crosslinked state. So it would be important to try to evaluate the crosslinks density values inherent to the materials obtained. Do these densities differ significantly for the four materials? How the authors have optimized the protocol of the crosslinking process? Could the crosslinks density values realized in the work be treated as optimal for practical use of the materials studied, or they need some further tuning?

Reply: Thanks for your comments. Cross-linked state of hydrogels can be evaluated by measuring mechanical properties or FTIR. Viscoelastic properties are useful methodology for studying chitosan hydrogels. Hydrogels are tridimensional networks with viscoelastic behavior (viscous and elastic behavior). Typically, the elastic modulus (G‘) is measured to understand the microstructure, heterogeneity, and cross-linked state of hydrogels. In other words, the storage modulus describes how much energy must be applied to distort the sample. If the state of crosslinked increases, then the rheological properties of the hydrogel should increased. Based on the elastic modulus, the samples (BC-H1, BC-H2, BC-H3, & BC-H4) displayed different viscoelastic behaviour. We described in the viscoelastic properties section that the elastic modulus of the chitosan hydrogels was higher than the loss modulus within the temperature range due to the formation of robust physical entanglement polymeric networks. The BC-H4 hydrogel was approximately 2.3 times folder than the BC-H1 and BC-H2 chitosan hydrogels and almost 1.5 times higher than the BC-H3 hydrogel at a constant temperature.

All hydrogels can be used in different biomedical applications. The selection of the hydrogel will depend on the degree of structuring or mechanical properties required. We believe that this methodology is useful because it allows us to obtain reproducible, sterile, non-toxic, and stable hydrogels that can be used in various biomedical applications.    

  1. The data presented show the substantial difference of the properties of crosslinked gel H2 and all other gels studied. H2 demonstrates very little intensity of diffraction peak centered at 2θ = 20° inherent to chitosan (Fig. 1b), and on the other hand is characterized by substantially larger pore size as compared to all other materials. Could the authors put forward some explanation of the “unusual” properties of this H2 hydrogel?

Reply: Thanks for your comments. The explication of the “unusual” properties of this H2 hydrogel were added.

  1. It would be reasonable to point out in the “Conclusions” section which of the four materials under study is more suitable for the practical use in biomedical applications.

Reply: As described above, all samples can be used in biomedical applications. We do not believe that one hydrogel is more useful than another; all have demonstrated excellent mechanical properties, wáter absorption properties, and biocompatibility.

Reviewer 2 Report

Please write an brief section related to a possible futuristic use of your chitosan cross linking in filed of drug delivery. Nice article. 

Author Response

Reviewer 2.

  1. Please write a brief section related to possible futuristic use of your chitosan cross-linking in the field of drug delivery. Nice article.

Reply: Thanks for your comments. The information was added to the introduction section.

Reviewer 3 Report

In this paper, the authors successfully produced chitosan hydrogels using sodium bicarbonate as a crosslinking agent at different concentrations and molecular weights. The goal is clear and the content is rich. However, there are still some issues to be addressed. The specific comments can be found as following

1.     Is there a problem with formula (2) in line 190?

2.     The figures in this paper are too vague, so the clarity should be improved.

3.     The numbering sequence in Figure 3 should be adjusted.

4.     Why authors applied chitosan in this work should be further clarified with more detailed introduction on the structure, properties and applications of chitosan with necessary supporting articles: Sources, production and commercial applications of fungal chitosan: A review; Recent advancements in applications of chitosan-based biomaterials for skin tissue engineering; etc.

5.     One scheme to show the experimental procedure is suggested for better understanding of this work to readers.

6.     All figures should be modified with a better readability, especially the quite small texts.

7.     Figure 6 is it necessary to add a ruler to the figure?

8.     Line 368 does not indent the first line of the paragraph.

9.     There are still some typos and grammar issues in the manuscript. Authors should carefully recheck the whole manuscript.

10. Authors should recheck the references to make sure full information is provided, such as volume, pages, etc. In addition, the format of references should be uniform.

Author Response

Reviewer 3.

  1. In this paper, the authors successfully produced chitosan hydrogels using sodium bicarbonate as a crosslinking agent at different concentrations and molecular weights. The goal is clear and the content is rich. However, there are still some issues to be addressed. The specific comments can be found as following:

Reply: The authors would like to thank the reviewers for providing critical comments and suggestions to improve the quality of the manuscript. We have revised the manuscript accordingly and included a detailed list of responses below. For your ease of reading, we have modified the manuscript in red.

  1. Is there a problem with formula (2) in line 190?

Reply: Thank you for your comments. The formula was corrected.

  1. The figures in this paper are too vague, so the clarity should be improved.

Reply: We apologize for the inconvenience. The quality of the Figures was improved.

  1. The numbering sequence in Figure 3 should be adjusted.

Reply: We apologize for the inconvenience. The numbering sequence in Figure 3 was adjusted.

  1. Why authors applied chitosan in this work should be further clarified with more detailed introduction on the structure, properties and applications of chitosan with necessary supporting articles: Sources, production and commercial applications of fungal chitosan: A review; Recent advancements in applications of chitosan-based biomaterials for skin tissue engineering; etc.

Reply: Thank you for your comments. The introduction was improved, and new references were added.

  1. One scheme to show the experimental procedure is suggested for better understanding of this work to readers.

Reply: Thank you for your comments. The scheme of the experimental procedure was added.

  1. All figures should be modified with a better readability, especially the quite small texts.

Reply: We apologize for the inconvenience. The quality of the Figures was improved.

  1. Figure 6 is it necessary to add a ruler to the figure?

Reply: Thank you for your comments. We added a ruler to Figure 6

  1. There are still some typos and grammar issues in the manuscript. Authors should carefully recheck the whole manuscript.

Reply: Thank you for your comments. The typos and grammar issues were corrected.

  1. Authors should recheck the references to make sure full information is provided, such as volume, pages, etc. In addition, the format of references should be uniform.

Reply: Thank you for your comments. The format of references was corrected. However, some recent articles (2023) do not have the information yet

Round 2

Reviewer 1 Report

The authors have presented the revised variant of the manuscript taking into account all the notes and recommendations of the reviewer. The article can be accepted for publication.

Author Response

Thank you for your comments.

Reviewer 3 Report

Authors have made proper revisions. However, one recent important paper on the structure, properties and applications of chitosan should be included: Sources, production and commercial applications of fungal chitosan: A review. After that, an acceptance is suggested.

Author Response

The authors would like to thank the reviewers for providing critical comments and suggestions to improve the quality of the manuscript. 

The suggested paper on the structure, properties, and applications of chitosan was included in the manuscript.